# Albumin Nanovectors in Cancer Therapy and Imaging

**DOI:** 10.3390/biom9060218

**Published:** 2019-06-05

**Authors:** Alessandro Parodi, Jiaxing Miao, Surinder M. Soond, Magdalena Rudzińska, Andrey A. Zamyatnin

**Affiliations:** 1Institute of Molecular Medicine, Sechenov First Moscow State Medical University, 119991 Moscow, Russia; surinder.soond@yandex.ru (S.M.S.); magdda.rudzinska@gmail.com (M.R.); 2Ohio State University, 410 W 10th Ave., Columbus, OH 43210, USA; jmiao3@hawk.iit.edu; 3Belozersky Institute of Physico-Chemical Biology, Lomonosov Moscow State University, 119992 Moscow, Russia

**Keywords:** albumin, nanomedicine, drug delivery, cancer

## Abstract

Albumin nanovectors represent one of the most promising carriers recently generated because of the cost-effectiveness of their fabrication, biocompatibility, safety, and versatility in delivering hydrophilic and hydrophobic therapeutics and diagnostic agents. In this review, we describe and discuss the recent advances in how this technology has been harnessed for drug delivery in cancer, evaluating the commonly used synthesis protocols and considering the key factors that determine the biological transport and the effectiveness of such technology. With this in mind, we highlight how clinical and experimental albumin-based delivery nanoplatforms may be designed for tackling tumor progression or improving the currently established diagnostic procedures.

## 1. Introduction

During the last decades, a large variety of carriers was generated from different organic and inorganic materials so as to encapsulate and enhance the delivery of very toxic and/or hydrophobic drugs, as well as to improve the sensitivity of the current diagnostic agents [1]. Metals [2], silicon [3], carbon [4], and polymers [5] represent only a fraction of the materials utilized at the nanoscale for achieving targeted drug delivery. Most of them showed high potential in pre-clinical investigations; however, biological materials like proteins or lipids have higher efficiency in the generation of translational nanotherapeutics [6]. The use of biological materials is advantageous for different reasons, including, but not limited to, high biocompatibility, lower toxicity, and reproducible large-scale fabrication. Albumin is one of the most easily purifiable natural products that can be nano-engineered, and it is a universal component of both liquid and solid tissues [7,8,9]. 

The clinical potential of albumin started to emerge in the last century, even though Hippocrates [10] described its properties, without knowing of its existence. Albumin clinical use was established during World War II [11], where it was used as a substitute agent for plasma, while in recent years, one of the first pieces of work describing the clinical use of this protein is dated 1944, describing its benefits in the treatment of cirrhosis [12]. 

Albumin is synthesized by the liver [13], and it is a small globular protein of about 5 nm with a molecular weight of 66.5 kDa. It represents the most abundant protein in the plasma, accounting for about 55% of the blood proteins (35–50 g/L in human serum). Albumin is a crucial player in maintaining colloidal osmotic pressure and regulating the plasma pH [14]. Also, it has a pivotal role in enhancing the bioavailability and regulating the transport of long chain fatty acids, nutrients [15], and metal ions [16], as well as of a variety of systemically administered pharmaceuticals, by increasing their bioavailability and stability in biological fluids [13]. 

Because of its natural properties as a blood transporter, the single molecule of albumin can be loaded and/or conjugated to different therapeutic payloads [17] in order to enhance their pharmacokinetics. For these reasons, Albumin was successfully tested to fabricate safe and cost-effective nanovectors [18], as they can be easily manufactured at the nanoscale, and they can accommodate an extremely versatile variety of the therapeutic, diagnostic, and theranostic payloads. In this review, we will focus on the recent advances in albumin nanovector (ANV) generation, providing insights into the synthetic processes, delivery properties, and applications in the treatment or diagnosis of cancer. 

## 2. Albumin as a Raw Material for Nanovector Generation

Known also as fraction V of the method optimized by Cohn [19], in order to purify serum proteins, albumin isolation is very consistent and reproducible, making this protein a cost-effective raw material for generating nanoparticles. Being an endogenous human protein, it is biocompatible, biodegradable, and by far non-toxic or immunogenic, even when the albumin nanostructure is assembled under harsh denaturation conditions and/or is cross-linked with potentially toxic agents. Its high biocompatibility can also be exploited for topical treatment in very delicate organs, like the eyes [20]. The only concerns raised against albumin-based nanomedicine are related to the risk of blood pathogens (e.g., virus and prions) contaminating the final product and derived from albumin purification. These concerns can be easily addressed by generating transgenic bacterial systems expressing this protein or through recombinant hosts like rice endospermine, where human albumin can be expressed efficiently [21]. Albumin has a highly conserved peptide sequence throughout mammals [22], and, in pre-clinical studies, it usually derives from human or bovine sources. The main difference between human and bovine albumin is that the former contains only one residue of tryptophan, while the latter contains two [23]. A pharmacokinetic comparison of albumin accumulation in the neoplastic lesions of Walker-256 carcinoma-bearing rats was performed using albumins derived from different sources [24], and no significant differences in tumor accumulation were detected. As a raw material for nanocarrier fabrication, albumin is very stable both under physiological conditions (its biological half-life is about three weeks) and in the presence of relatively high concentrations of solvents or heterogeneous pHs. Despite the high number of studies dedicated to albumin carrier properties, it was only in the 1990s (thanks to the pioneering works of Dr. Carter and Dr. Curry) that these features were elucidated through crystallography studies [25,26].

Human albumin has three domains (I–III) with a similar tridimensional structure [27], and each of these domains is composed of two sub-domains (A and B). The drug binding properties result from two specific sites [28], named site I (known as warfarin binding site) [27] and site II (known as benzodiazepine binding site) [27], located in domain II and III, which are able to bind different molecules through hydrophobic and hydrophobic/electrostatic interactions, respectively. However, while these protein sites have been identified and extensively characterized, many pieces of evidence indicate that other less characterized protein binding sites could participate in the molecular drug interaction [26,29]. For instance, the existence of a third drug binding site in domain I have been proposed [30]—in Figure 1, the albumin structure and its binding sites are shown. 

Furthermore, the albumin structure is characterized by a high number of free amino and carboxylic groups [31], which allow for covalent or non-covalent modifications based on amide condensation, ionic interactions, and hydrophobic adsorption. Many functionalization protocols are based on a thiol group provided by cysteine at position 34 [32], located in a small pocket in domain I. Also, albumin possesses excellent hydration properties [33], and it can be used with other organic and inorganic nanomaterials in order to increase their stability and their biocompatibility. Albumin can be functionally optimized with standard genetic engineering procedures, allowing for the development of a final product with particular features, such as additional thiol groups for further covalent modification [34]. Albumin binding was shown to protect the payload against reactive oxygen species (ROS) and degradation in biological fluids [35]. Finally, Albumin is pH-stable (in a range of 4–9) and is thermo-resistant (stable for at least 10 h at 60 °C), representing an optimum substrate for various synthetic processes and chemical syntheses, while it can be stored, with minor changes in its structure, at relative high concentrations (5–20%) for many years [36].

## 3. Synthesis of Albumin Nanovectors

Multiple protocols are currently available for generating ANVs [37], even though two main synthesis routes are used in the field, namely: (1) high-pressure homogenization and (2) desolvation followed by cross-linking. High-pressure homogenization is the method used to generate the Food Drug Administration (FDA) approved therapeutic Abraxane^®^ (known as albumin-bound (nab) technology), and it has been shown to be extremely efficient for the encapsulation of hydrophobic payloads like chemotherapeutics (for example, taxols). High-pressure homogenization is purely a mechanical procedure used to generate nanoparticles by forcing a fluid through a very narrow gap under high-pressure. The system is composed of a pump and a nozzle [38] and can be applied to a crude emulsion (constituted by albumin, the payload, and the solvents necessary to solubilize the payload). The pressures are applied in a range between 50 and 500 MPa—traditional homogenization is usually achieved at pressure levels below 50 MPa. During high-pressure homogenization, the fluid undergoes high shear stress because of the passage through a restrictive valve, resulting in the formation of nanodroplets [39] (Figure 2A). The mechanical energy applied to the system can generate heat; therefore, the homogenizers are usually equipped with refrigerating systems to control the temperatures of the process, a necessary step when the payload is thermolabile [40]. However, if a high-pressure homogenizer is not available, many publications indicate using ultrasonication as an optimal method in order to induce albumin aggregation. From a molecular standpoint, high-pressure homogenization can affect the tertiary and the quaternary structure of globular proteins, leading to protein unfolding [41]. 

Compared to the aggregation state of equine albumin induced with a high temperature, protein aggregates obtained with high pressure are reversible, and no formation of intermolecular β-sheets was detected [42]. The albumin amphipathic properties favor nanoparticle self-assembly through the occurrence of weak chemical bonds based on Van der Waals and ionic interactions [43,44], which preserve the biological properties of this protein. However, ANVs generated through this procedure have limited stability, in particular when injected intravenously, where the nanoparticles disaggregate as albumin-bound drug molecules upon interaction with endogenous circulating albumin [45].

Desolvation is a process where the albumin molecules aggregate when exposed to a solvent such as ethanol. In this case, the solvent induces the displacement of the water molecules, affecting the overall secondary structure of the albumin. In brief, the solvent can turn albumin β-sheets into α-helices, while breaking intramolecular hydrogen bonds within the β-sheets of the protein and inducing new intermolecular hydrogen interactions [46]. Protein denaturation dynamics are governed by general physical and chemical parameters, like the pH [47], temperature [48], and the kind of the solvent. The system is usually then stabilized via cross-linkers (i.e., glutaraldehyde or 1-ethyl-3-(3-dimethylaminopropyl)carbodiimide; Figure 2B). After chemical stabilization, the unreacted cross-linker is washed away, reducing any potential cytotoxic effect when administered in vitro or in vivo. The cross-linking with glutaraldehyde links [49] the available amino groups of the protein (lysine and arginine residues) [45], while at the same time limiting further surface modifications involving these reactive groups. One shortcut to avoid this phenomenon is to pre-conjugate albumin with dimethylmaleic anhydride to protect some amino-groups during the cross-linking step [50]. Alternatively, cross-linking can also be achieved through electrostatic interactions by using natural molecules like chitosan [51]. 

Chemical cross-linking allows for the formation of a very stable final product, even within the circulatory system, but this procedure can affect the overall trafficking of the particles, favoring their sequestration by the mononuclear phagocytic system (MPS). 

Recently, the group of Dr. Cui performed an investigation devoted to evaluating the pharmacokinetic advantages of using cross-linked ANVs over non-cross-linked ANVs [45], where the particles were designed to deliver paclitaxel (PXT) in an in vivo model of sub-cutaneous prostate cancer. What was interesting was that the study did not reveal any significant difference in PXT clearance between the two ANV formulations, despite the theoretical higher stability expected by the cross-linked ANV platform. The authors justified this phenomenon, claiming that the pharmacokinetics of crosslinked ANVs was affected by (1) the higher sequestration of the nanotherapeutics by the element of the MPS, and (2) the exchange of the drug from the particles to the free circulating albumin. Surprisingly, the non-cross-linked nanoparticles showed the highest antiproliferative effect, while no significant differences in particle toxicity were registered. These data seem to confirm that the use of cross-linkers can change the trafficking of the nanoparticles, and that, at least in the case of PXT, the use of ANVs generated through homogenization is more advantageous.

## 4. Albumin and Albumin Nanovector Receptors 

Albumin physiologic tissue and cellular depends on at least seven specific receptors that are very heterogeneous in function, tissue distribution, and cell phenotype expression [52]. Gp60 and secreted protein acidic and rich in cysteine (SPARC) receptors are the most important because they were shown to be involved in the transport of ANVs in cancer diseases. Dr. Schnitzer discovered that Gp60 (known as albondin [53]) is expressed in the continuous endothelium, and has a very high affinity for albumin [54]. The group of Dr. Malik elucidated this interaction, unveiling the role of this receptor in the transcytosis of albumin via caveolae formation [55]. Upon interaction with the Gp60 receptor, albumin induces the phosphorylation of caveolin-1, a crucial step in caveolae formation, through the activation of the Src kinase signaling pathway [56]. Caveolin trafficking mediated by Gp60 was further associated with the activation of endothelial nitric oxide synthase and nitric oxide production, regulating the vascular tone. More recently, it was shown that this receptor is also expressed in the pulmonary epithelium, where it mediates albumin trans-alveolar transport [57]. From the standpoint of drug delivery, this receptor is not only important because it can mediate ANV transport over two major biological barriers (epithelial and endothelial barrier), but also because it is overexpressed on cancer cells, potentially functioning as a targetable molecule [58]. Recently, it has been hypothesized that it also favors the uptake of inorganic nanoparticles coated with albumin [59].

Also known as antiadhesin, osteonectin, BM-40, and 43K protein, SPARC is a protein secreted by many cellular phenotypes, and it can interact with both the cell surface and with the extracellular matrix, thus inhibiting cell adhesion [60]. This receptor has a pivotal role in embryonic development, and, in the adult organisms, is expressed in high turn-over tissues [61], as well as during injuries and pathological conditions [62]. It was shown that SPARC can modulate the effect of different growth factors, and its activation was correlated with G1 cell cycle arrest [62]. Its structure is very similar to Gp60, even though significant differences in the N-terminal region of these receptors have been identified [55]. Its expression is associated with pathophysiological conditions involving extracellular matrix remodeling, including cancer [63] and neoangiogenesis processes [64]. Also, SPARC was shown to affect the endothelial cell morphology and vascular permeability probably via the modulation of the F-actin expression, which in turn increases the vascular intercellular gap formation and paracellular extravasation of the macromolecules [65]. Its over-expression in cancer diseases was proposed as a prognostic tool, and it was correlated with the efficacy of ANVs in inhibiting cancer proliferation [66]. Recently, a mannosylated albumin delivery platform able to target both SPARC and CD206 was proposed in order to provide a more efficient way to target drug-resistant cancer cells and reprogram tumor-associated macrophages that over-express the mannose receptor [67]. 

Gp30 and Gp18 are scavenger receptors characterized by a high affinity for damaged (not enzymatic glycosylation, oxidation, and fixation) albumin [68,69]. These receptors are expressed in different cells and are involved in the endo-lysosomal sequestration and catabolism of this protein [70]. Despite the high number of citations referring to these receptors as being implicated in albumin and ANV trafficking, we were not able to find a direct link between their interaction with ANVs and drug delivery [71]. 

As soon as nanoparticles are injected intravenously, they are quickly opsonized by a plethora of circulating molecules (including IgG), as they are recognized as non-self-elements. This phenomenon (known as protein corona) favors particle internalization in phagocytic cells via interactions with Fc-γ receptors [72,73], which are expressed ubiquitously on the surface of the cells of the immune system, and recognize the Fc of IgG favoring the internalization and degradation of antibody-opsonized agents. In this scenario, ANVs composed by denaturated albumin could adsorb the circulating IgG and be internalized through Fc-γ receptors, and eventually be sequestered to the endolysosomal compartment. 

Alternatively, the neonatal Fc receptor for IgG (FcRn, also known as the Brambell receptor) can determine a different fate for ANVs after internalization. FcRn is a transmembrane protein composed of a b2-microglobulin (B2M) and an α-chain of the MHC class I complex. This protein was very well characterized for its role in rescuing the IgG of the maternal source, and in transferring passive humoral immunity from the mother to the fetus via the syncytiotrophoblasts of the placenta, or from the milk via the enterocytes of the duodenum. The efficient targeting of this receptor was recently indicated to be pivotal for the future generation of ANVs designed for oral delivery, as it can be expressed in the intestine epithelium [74]. Most importantly, FcRn allows for albumin recycling after internalization, protecting it from endolysosomal degradation, which is eventually responsible for the reintroduction of this protein into circulation [75]. Mice lacking this receptor showed a higher degradation of albumin with a consequent decrease of its half-life, indicating that FCRn receptors can rescue albumin from degradation, in addition to its payload. This receptor is expressed in many tissues, such as gastric and renal epithelium [76]) and in different phagocytic cells [77], and it is under surveillance by the scientific community for the development of more effective vaccines [78] that may also be able to overcome epithelial biological barriers [79]. A schematic showing the receptors potentially involved in ANV trafficking and degradation is shown in Figure 3.

## 5. Albumin Nanovectors in Cancer: Lesson Learned from Abraxane^®^

As for many other delivery platforms, most of the literature dedicated to ANVs is focused on improving the delivery of chemotherapeutics, as these drugs showed a poor availability, low aqueous solubility, and high toxicity. A significant increase in the number of the papers published in the field occurred after Abraxane^®^, the prototypical model for albumin-based nanotherapeutics, and one of the most successful examples of clinically translated nanotherapeutics (Figure 4). 

This technology is favored is massively tested in clinics, and the recent ongoing and terminated clinical trials based on ANVs are shown in Table 1. Today, Abraxane^®^ is approved for the treatment of different oncologic diseases, including metastatic breast cancer, non-small-cell lung cancer (NSCLC), and, more recently, metastatic pancreatic cancer. Abraxane^®^ (known as ABI 007 or nanoparticle albumin-bound (nab)-PXT) completely revolutionized the delivery of PXT for which the clinical formulation was traditionally based on cytotoxic solvents like Cremophor^®^ EL and ethanol, which are associated with severe side effects (including anaphylaxis). 

Abraxane^®^ consists of PXT-loaded human-derived ANVs (130 nm in size) assembled through high-pressure homogenization. The drug finds its accommodation in the albumin protein at the interface of subdomains IIA and IIIA and between domains I and III [80]. Its stability is enhanced by weak hydrophobic bonds [14] between the proteins and the therapeutic, encapsulated in the particle structure. The Abraxane^®^ mode of action exploits different pathways characterized by tumor vasculature and cancer albumin transport. Firstly, Abraxane^®^ is internalized in endothelial cells through the Gp60 receptor and is transported into the interstitial tumor space via caveolin-mediated transcytosis [81]. This active transport mechanism favors the overall increase in PXT penetration into the biological tissues, in particular, in cancer lesions, where the nanotherapeutic is highly retained [82,83]. Less information about the cancer cell uptake is available, even though other albumin receptors could participate in this phenomenon [84]. Of note, the tumor sensitivity to Abraxane^®^ was positively correlated with the overexpression of SPARC [85]. Interestingly, a cell line of non-small cell lung cancer (a variant of A549 cells) was found, via proteomic analysis, to be resistant to Abraxane^®^ [86], highlighting the differential protein expression profiles between the sensitive and non-sensitive cell lines to the nanotherapeutic. However, these proteins were not related to the known biomolecular mechanisms of resistance to PXT, indicating the possibility that Abraxane^®^ could generate a unique uncharacterized mechanism of resistance. While the therapeutic advantages of Abraxane^®^ over Cremophor^®^ EL formulation are not in discussion, the drug encapsulation in these particles did not result in an increased pharmacokinetic profile and the serum half-life of PXT [87]. The drug retention was shown to be consistent in aqueous media within the galenic formulation of the therapeutic, but in the presence of serum, the stability of the system decreased, probably due to the therapeutic exchange phenomena between the nanoparticles and the free albumin molecules. A new formulation of albumin conjugated with cholesterol was designed to generate carriers with higher stability in the blood environment, as well as to increase the encapsulation efficiency of the drug [88]. In this case, the system was generated by conjugating the cholesterol to free albumin via a succinimidyl bond, and the particles were assembled and loaded through ultra-sonication at 4° in the presence of the chemotherapeutic. Compared to the traditional formulation, the use of cholesterol decreased the PXT release by two-fold, while increasing particle colloidal stability and internalization in cancer cells. Also, cholesterol modification increased the serum half-life of the drug, showing enhanced cytostatic properties in a preclinical model of melanoma. The same engineering principles used to encapsulate PXT in Abraxane^®^ can be theoretically applied to encapsulate many other lipophilic payloads. Many chemotherapeutics, because of their hydrophobic/amphipathic structure, can easily find non-covalent accommodation in the hydrophobic molecular pockets characterizing the albumin backbone. Following these principles, ANVs were designed to deliver fenretinide to improve current treatments for NSCLC. This molecule belongs to the category of retinoids, but when freely administered, it is characterized by its poor bioavailability [89]. After encapsulation in ANVs, fenretinide increased its anti-proliferative effects in vivo, and it more effectively killed three-dimensional (3D) spheroids of lung cancer cells than free drug [36]. This evidence was supported by detecting caveolin-1 (a key player in caveolae formation) over-expression in the cancer cells, and hypothesizing that an active transport (similar to albumin transcytosis in endothelial cells) is active directly within the spheroids’ mass. However, such speculation was not supported by the detection of any receptor associated with the transcytosis pathway, and more investigations are necessary. 

Albendazole (FDA approved as an anti-anthelminthic drug) was encapsulated in ANVs so as to exert its anti-angiogenic properties towards a xenograft model of ovarian cancer [90]. Interestingly, the authors demonstrated that the particle size was dependent on the ratio between albendazole and albumin, as ratios of 1/100 and 1/5 generated 10 and 100 nm particles, respectively. However, both kinds of particles were characterized by a controlled release of the drug occurring for eight days after an initial burst release (in which 35–50% of the drug was released) in the first six hours. Despite the theoretical anti-angiogenic properties of albendazole (demonstrated through the efficacy of both the formulations in decreasing the extent of ascites in the xenograft tumors), the particles (in particular the 10 nm carriers) showed pronounced toxicity towards ovarian cancer cells, while no significant toxicity was registered in the normal cells. Encapsulation in ANVs was efficient in decreasing the side-effects of the second-generation tyrosine kinase inhibitor, dasatinib [91], which is FDA approved for chronic myeloid leukemia and Philadelphia chromosome-positive acute lymphoblastic leukemia. However, dasatinib treatment can be restricted or discontinued because of the occurrence of severe edema and pleural effusion [92]. This phenomenon is due to the inhibitory action of dasatinib on Lyn, a tyrosine kinase that regulates the maintenance of the vascular barrier integrity. ANVs loaded with this drug showed higher cytostatic activity towards K562 leukemia cells, and, more importantly, increased vascular protection towards endothelial cells. ANVs were also tested to encapsulate gemcitabine (FDA approved for the treatment for pancreatic cancer) [93] to increase its in vivo half-life, as this drug is very hydrophilic and unstable when systemically administered. Encapsulation in ANVs allowed for a prolonged release of about five days, after an initial burst release of less than 20% during the first hour after administration. The particles showed increased tumor toxicity in vivo, with no evident side-effects, highlighting the versatility of this technology in encapsulating hydrophilic drugs.

## 6. Targeted and Complex Albumin Nanovectors

The relative abundance of reactive sites on the ANV surface can allow for the functionalization of multiple targeting moieties and complex synthesis designs. ANVs were designed to target hepatocarcinoma encapsulating 10-hydroxycamptothecin (an analog of the topoisomerase I enzyme inhibitor camptothecin) via covalent surface modifications with glycyrrhizic acid (a molecule purified from the licorice plant), as liver cancer cells over-express the receptor for this molecule [94]. Excellent targeting properties towards colon cancer were achieved by modifying the surface of the particles with both tumor necrosis factor-related apoptosis inducing ligand (TRAIL) and transferrin [50]. Despite its targeting action, TRAIL can also induce cell apoptosis by binding two specific death receptors over-expressed in cancer cells (DR4 and DR5). The system was also further encapsulated with doxorubicin (DOX), in order to guarantee high cytostatic power. Transferrin, in particular, was shown to increase cell sensitization against DOX, providing an ultimate tool to overcome tumor drug resistance. The versatility of ANVs in surface functionalization was tested with many kinds of targeting molecules, including antibodies like trastuzumab [95] (FDA approved for HER2+ breast cancer), cetuximab [96] (FDA approved for colon cancer), apolipoproteins [97], nanobodies (antibodies lacking the Fc domain [98]), hormones [99], and biotin [100]. Moreover, multiple drug encapsulation can combine therapeutic and MDR inhibitors, as in the case of albumin nanoparticles loaded with docetaxel (chemotherapeutic) and quercetin (P-Gp efflux pump inhibitor), respectively [101]. 

These carriers can also be designed to induce the payload release as a function of the physiological pH. These smart ANVs permitted a fast release of mitoxantrone (chemotherapeutic) at an acidic pH and a slower release at neutral or basic pHs. This system was conceived to target the tumor microenvironment (which compared to healthy tissue is relatively acidic) and to maximize the release of the drug when the carrier is internalized into the cellular endo-lysosomal vesicles [102], which are acidic in pH. The release mechanism is based on the pH sensitivity of the coordination bonds between the albumin and the zinc ions that are used to stabilize the structure of the nanoparticles [103]. A typical desolvation process, which combined zinc absorption with the albumin particles (exploiting two specific sites present in the structure of the protein for this metal [104,105]), was used to generate the carriers. The bond between zinc and albumin weakens at a low pH, favoring the release of the drug stabilized by the metal (Figure 5). 

Alternatively, the group of Dr. Zhu generated a system composed of two kinds of albumins. One albumin was modified upon covalently binding doxorubicin through a peptide sensitive to cathepsin B to release the drug after particle internalization in the lysosomes, and the other one was modified with the peptide K237 to provide efficient targeting towards the tumor vasculature. More importantly, the method used to assemble the particles was based on celecoxib, an FDA molecule used to target inflammation via Cox-2 inhibition. This formulation increased the antiproliferative effects of DOX, while providing an anti-inflammatory effect, and, above all, a novel way to stabilize albumin nanoparticles with no chemical cross-linking [106]. When doped with collagenase, ANVs were also tested to digest the tumor extracellular matrix, and favored the tumor penetration of the active principle riluzole (an inhibitor of glutamate receptors, FDA approved for amyotrophic lateral sclerosis) and curcumin (an inhibitor of NFkB pathway [107]). The tumor penetration activity and killing were measured on melanoma spheroids, and the particles coated with collagenase showed a higher concentration in the internal region of the cellular spheres [107]. 

While albumin in healthy individuals cannot overcome the blood–brain barrier, different brain tumors, like gliomas, can enhance its uptake, as they can represent an optimal fuel for increasing amino-acids intake and energy [108]. The receptors (overexpressed in glioma) for the albumin were identified to be Gp60 (over-expressed in the endothelium that in turn, increase extravascular leakage of Albumin) and SPARC [109,110]. To this purpose, the ANVs were loaded with both PXT and Fenretinide so as to achieve the maximum cytostatic activity on the glioma growth. The particles were produced by using a desolvation method based on the reducing agent NaBH4, in the presence of high concentrations of urea, and further stabilization was achieved by loading the two hydrophobic drugs. Even though the albumin was denaturated, the system was still able to colocalize with SPARC in vivo, which is known to interact with native albumin. Here, the tumor growth inhibitory properties of the system were verified both in the subcutaneous and intracranial models of the disease [111], significantly highlighting the importance of Gp60 and SPARC in ANV transport, also when produced via the desolvation method. Recently, the development of cationic carriers was proposed to delivery nucleic acids exploiting ionic interactions between the carriers and the payload. However, without a proper coating, their injection results in fast opsonization and clearance in the MPS. To this end, the albumin structure can be modified with cationic groups like ethylenediamine [112] to facilitate the loading of the siRNA molecules. Such a system was able to mediate the cytoplasmic delivery of siRNA, as its high content in thee amine group induced the endolysosomal escape of the nanoparticles by destabilizing the endosomal membrane. 

## 7. Albumin Nanovectors as Diagnostic and Theranostic Agents

Radiolabelled, colloidal albumin aggregates are currently used to detect sentinel lymph nodes in tumor diseases. The most investigated diagnostic platform is called nanocoll, which is formed by small particles (mean diameter = 8 nm) [113] of human serum albumin radiolabeled with Technetium 99 m (half-life of about six hours), and it was used in the clinic for the last 30 years [114]. This system is specifically designed to perform lymphoscintigraphy evaluating lymphatic integrity, obstruction, and lymph node malignant infiltration of several kinds of cancers, by injecting the particles intradermally in proximity to the neoplastic lesions [115]. Recently, van der Poel et al. [116] included indocyanine green in the system to increase its sensitivity by integrating radioactive and fluorescent imaging during a robotic-assisted laparoscopic prostatectomy. Encapsulation in ANVs was shown to increase the diagnostic properties of near-infrared dyes, as the encapsulation in the ANVs of a derivative form of IR-783 was shown to enhance the fluorescent imaging in a model of colon cancer [117]. In addition to traditional chemotherapeutics, ANV versatility allows for the functionalization/loading of many kinds of fluorescent or paramagnetic probes for the development of biosensor applications based on detectable fluorescent signals [118]. Some magnetic and near-infrared agents can function either as imaging agents or as therapeutic (killing) agents. These probes can absorb irradiated external energy (e.g., magnetic field or infrared light) and convert it into a detectable signal and heat [119,120], offering novel oncological treatments based on thermoablation. Also, this strategy allows for the detection and the localization of the particles within the body, opening new avenues for the development of multifunctional nanomedicine approaches, combining diagnostic and therapeutic properties (theranostic). Indocyanine green represents the prototypical example of a theranostic agent, whose efficiency is strictly dependent on its encapsulation, as it is characterized by low solubility and stability in biological fluids. Upon proper stimulation with infrared light, both photodynamic and photothermal therapy [121] can be achieved by increasing the oxidative stress and the temperature of the surrounding environment, respectively. The encapsulation of indocyanine green within ANVs also increased its uptake within cancer cells, as well as its killing properties. Many scientists are currently focusing their efforts on generating a single delivery system combining photothermal and chemotherapy. In support of this, mild heating was shown to increase the cellular penetration of the drugs (by enhancing membrane permeability), as well as inducing tumor cell autophagocytosis [122]. Also, heat generation can accelerate the release of the therapeutic payloads, providing new opportunities for the development of tools able to induce a controlled release. To this purpose, Abraxane^®^ was combined with indocyanine green, as the use of two FDA approved molecules [123] could favor the further translational development of this system. The presence of PXT in the nanoparticles also promoted the stabilization of the NIR dye in the system, both in vitro and in vivo, resulting in an enhanced circulation half-life and cytostatic properties, probably mediated by SPARC targeting [124]. Additionally, Chlorine e6 (Ce6) was also extensively investigated to develop ANVs with theranostic features [125], as the hydrophobic character of this molecule favors ANV self-assembly. When illuminated with LED light, this molecule induced the formation of ROS formation, while, at the same time, allowing for ANV fluorescence detection and quantification. 

Multifunctional ANVs with targeting, and chemo- and photo-therapeutic properties were recently developed by modifying the albumin molecules with cyclic RGD (targeting tumor vasculature) in combination with Ce6. The particles were assembled by following a one-step procedure in which modified albumin molecules with RGD or Ce6 are mixed in the presence of PXT, or by initially generating a core of albumin–Ce6–PXT, followed by a coating of albumin–RGD. In both of these cases, the payload of PXT served as a driver for particle auto-assembly, giving a stable multi-therapeutic nanoparticle model [126]. Both fluorescence and magnetic resonance imaging could detect these particles, and upon irradiation, they could affect the stability of the endosomal compartment, as well as releasing the drug, providing synergistic anti-proliferative effects on the targeted cells. Similarly, ANVs loaded with croconine showed both theranostic and biosensor properties. This molecule has two different peaks of absorption between a basic (680 nm) and acid pH (790 nm), which can be used to detect the surrounding acidity. Croconine was loaded in the albumin molecules through hydrophobic bounds, and, like Ce6, it induced nanoparticle self-assembly. ANVs loaded with croconine were able to reveal the pH of different areas of the tumor lesions through photoacoustic imaging [127], even when applied to large tumor volumes (200 mm^3^). ANVs can also be modified with the contrast agent gadolinium (Gd) by exploiting the Gd-diethylenetriaminepentaacetic acid (DTPA) that has chelation properties for this element. After encapsulation, Gd showed an increased residence time in the body and a higher cancer tissue specificity [128]. The system was successfully tested in vitro and in vivo for its ability to target and detect hepatic carcinoma, showing enhanced contrasting properties in the T1-weighted magnetic resonance imaging (MRI), and a similar approach resulted in being efficient for improving the MRI of the brain [129]. 

These particles are also suitable for the development of image-guided therapy, to follow the pharmacokinetics of the payload independently from the carriers. In this case, ANVs were loaded with rapamycin and modified with the fluorescent molecule Cy5. The delivery of rapamycin was detectable by co-transfecting the head and neck carcinoma cell line HN12 with the N- and C-terminal fragments of the firefly luciferase. Rapamycin could induce the dimerization of these two fragments and consequently bioluminescence when delivered within these cells. Through this model, it was demonstrated that while the ANVs reached a peak of accumulation in the tumor 24 h after nanoparticle injection, the highest level of bio-luminescence (corresponding to tumor accumulation of rapamycin) was at six hours. This discrepancy probably depended on the action of the drug inhibiting protein translation six hours after injection, including the luciferase fragments’ expression that was responsible for the generation of bioluminescence. 

## 8. Harnessing Endogenous Albumin and Hybrid Systems

After intravenous injection, the occurrence of a protein corona on the surface of the nanoparticles is a very well documented phenomenon that eventually can change the delivery and the targeting properties of the carriers, favoring their sequestration in the MPS [130]. As albumin is the major protein component of the serum, it is usually involved in this phenomenon. On the other hand, supporting the formation of a pure albumin corona on the particle surface can increase the circulation properties of the carriers, as well as the targeting of Gp60 and SPARCS. To this goal, an albumin-binding domain was efficiently used to functionalize PXT-loaded micelles, increasing their cancer accumulation and cytostatic effect in an in vitro and in vivo model of breast cancer [131]. The group of Dr. Yoon developed a theranostic tool able, in the presence of albumin [132], to exploit the avidity, of the tumor tissue for this protein. The system is composed of nanovesicles of phthalocyanine that in an aqueous solution have self-assembly in nanostructures, but when injected, disassemble to bind albumin, generating ROS and infrared fluorescence. Endogenous albumin was also harnessed to develop a more effective cancer vaccine exploiting the long circulation time of this protein, as well as its continuous perfusion of the lymph nodes [133]. The system was based on Evans blue conjugated with different adjuvants and antigens to exploit the high affinity of this dye for circulating albumin. 

On the other hand, albumin can be used as a functional component in the synthesis of nanoparticles, as it was shown to increase nanoparticle dispersity and stability while adding a new layer of loadable modifiable material [134]. The adsorption of albumin on the surface of nanoparticles occurs naturally through hydrophobic and electrostatic bonds, because of the amphipathic character of this protein. However, some authors report that emulsion-solvent evaporation and microemulsion techniques provide a more efficient formation of albumin coating [135]. Most of the delivery platforms investigated so far can interact with albumin in their pristine form, including metallic, magnetic, lipid, polymeric, and carbon carriers. Generally, the adsorption of albumin can increase their size significantly and change their surface charge, in particular, in the case of positively charged particles [136]. More importantly, the presence of albumin can affect the release properties of the carriers, as the payload can be adsorbed by this kind of coating [137]. An albumin shell on the nanoparticle surface was shown to decrease their accumulation in the MPS and their internalization in the macrophages, as it can inhibit the interaction with some serum proteins like IgG [138]. Also, albumin can increase nanocarrier biocompatibility. Albumin-coated silica nanoparticles showed decreased red blood cell lysis properties [139], while when this protein was adsorbed on the surface of multiwall carbon nanotubes, it reduced the platelet aggregation and ROS production induced by these nanoparticles [140]. Albumin coating increased the yield of antisense oligonucleotides and siRNA transfection when used to coat positively charged liposomes [141] while decreasing their toxic effects. Similarly, Zhang et al. [142] combined PEI-Fe3O4 magnetic nanoparticles with albumin via standard desolvation/cross-linking protocols to increase their biocompatibility, delivering more efficiently a plasmid to overexpress Interferon-γ in cancer cells. Figure 6 is a schematic representing the advantages of a surface functionalization based on albumin. 

## 9. Inhalable Albumin Nanovectors

Albumin is significantly present in the fluids lining the lungs [143], although at a lower concentration than in the blood. In this organ, it physiologically contributes to tissue transport and metabolism, and for this reason, ANVs were designed for aerosol inhalation, considering the low impact that this material has on the pulmonary organ. Recently, Woods et al. [144] demonstrated that the inhalation of albumin nanoparticles does not induce any significant side effects. Only at the higher dose used (16 mg/kg), did the particles produce a mild increase in inflammatory cytokines and immune cell infiltration. More importantly, they demonstrated that the ANVs (generated via desolvation/crosslinking method) could significantly reside in the lung for at least 48 h before being cleared or transported to other organs, overcoming the mucosal barrier. When conjugated with DOX, ANVs were stabilized with octyl aldehyde, which allowed for particle generation without the use of any potential toxic cross-linker. The particles were eventually adsorbed on their surface with TRAIL, which was shown to synergize with chemotherapeutics and to increase the cytostatic effects of the nanocarriers while having a mild impact on healthy cells [145]. This technology had a final size of about 350 nm, and the particles were administered via aerosolization to favor their pulmonary targeting. The carriers showed promising local antitumor effects against lung cancer, even though detectable tumor shrinking was registered 72 h after treatment, probably because of the slow release of the therapeutics. However, it was shown that in order to achieve a higher lung deposition via inhalation, the best size for the particles is in the microscale (from 1 to 5 µM) [146].

Chaurasiya et al. [147] tested different sizes of albumin particles (0.5, 1, and 3 µM) to generate homogenous 5 µM dry powder via a spray-drying technique. Their goal was to evaluate whether the initial size of the particles could affect the final residence time and the cytostatic effect of the inhalable dry powder. They showed that the dry powder generated with the smaller particles had a faster pulmonary clearance and lower efficacy in killing lung cancer via PXT release, demonstrating that the initial size of the albumin aggregates is fundamental for generating efficient inhalable technology. 

## 10. Conclusions

The encapsulation of pharmaceuticals in ANVs or albumin coatings has been independently demonstrated several times to decrease the cytotoxicity of the payload or the coated carriers [148] while increasing their water solubility, bioavailability, and protecting them from the insults of the biological environment. Albumin versatility in accommodating different payloads and in being modified with simple “click-chemistry” methods, make this protein an eligible candidate for developing universal nanomedicine to treat cancer and other pathological conditions. Many works attribute the clinical potential of albumin and ANVs to albumin receptors, highlighting their key role in the mechanisms and/or the biological trafficking processes central to this technology. To this end, Gp60, SPARC, and FcRn targeting should be monitored in applications using ANVs synthesized without protein denaturation, while Gp30, Gp18, and FcRγ could be activated when ANVs are fabricated through denaturing methods. Alternatively, while more studies on the involvement of Gp 30 and Gp18 in the transport of ANVs are necessary, some findings indicate that modified albumin can also interact with Gp60 and SPARC, posing an urgent need for more research in this direction to verify these possibilities. 

Many aspects related to ANV synthesis and loading have yet still to be clarified—as it is not clear when (in the case of the accommodation of hydrophobic payloads) high-pressure homogenization is necessary to generate the particles, and when particle self-assembly procedures can be sufficient to produce a stable product instead. Also, while crosslinking with traditional fixatives like glutaraldehyde guarantees a high reproducibility of the final product, other methods involving simple exsiccation and re-hydration, or the use of other biological molecules (e.g., hyaluronic acid or chitosan) can be used for the same purposes, thus preserving the function of biological payloads. Finally, we believe that more investigations have to be performed to understand the overall biological impact of using albumin nanoparticles on the whole organism. Albumin accumulation in the extravascular space is a common characteristic of many pathological conditions, like cancer, infections, and immune disorders [14], and many of these diseases are characterized by cachexia [30,149,150]. Albumin breakdown in the affected tissue was indicated as a possible contributor to this life-threatening condition. In the case of cancer, for example, it is well known that interstitial albumin is used as a source of biological energy by the tumor [149,150]. Dr. Kratz reported that albuminemia is probably (together with Gp60 and SPARC) one of the targeting mechanisms at the base of the Abraxane success, as cancer cells can avidly consume this protein [151]. In this scenario, it would be interesting to understand whether the accumulation of injected ANVs to the pathological site could mitigate this phenomenon or exacerbate tumor progression.

## Figures and Tables

**Figure 1 biomolecules-09-00218-f001:**
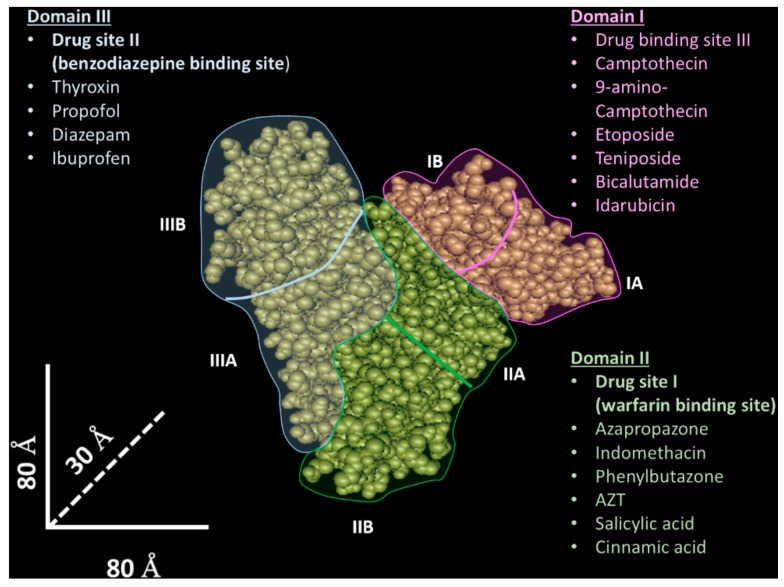
Tridimensional albumin structure and drug binding sites. The albumin tridimensional structure (source National Center for Biotechnology Information- https://www.ncbi.nlm.nih.gov/) is composed of three domains (I–III) highlighted with different colors (light blue—domain I; green—domain II; pink—domain III). Other binding sites interacting with ions, small molecules, and peptides are not shown.

**Figure 2 biomolecules-09-00218-f002:**
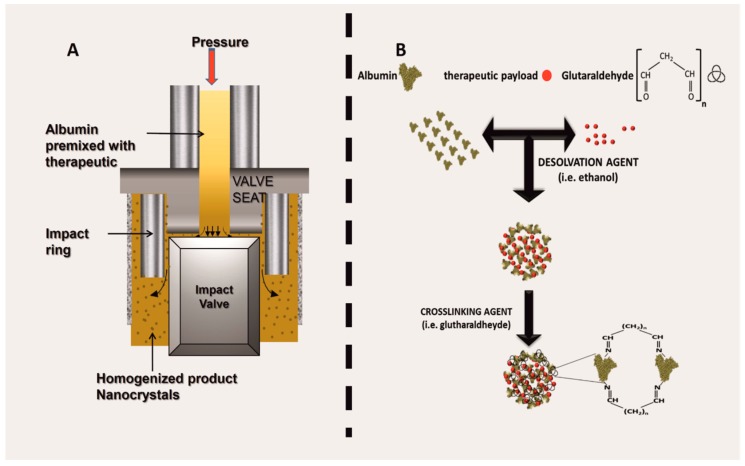
Albumin nanovector (ANV) synthetic routes, namely: (**A**) schematic of the working mechanism of a high-pressure homogenization; (**B**) synthesis through desolvation, followed by crosslinking.

**Figure 3 biomolecules-09-00218-f003:**
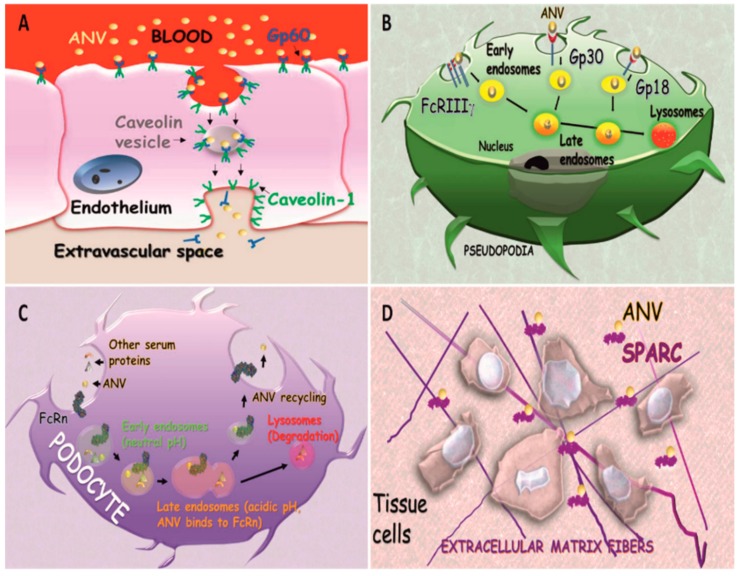
Albumin receptors, as follows: (**A**) transendothelial transport of ANVs mediated by Gp60; (**B**) trafficking and endosomal degradation of ANVs mediated by FcRIIIγ, Gp30, and Gp18 in phagocytic cells; (**C**) Albumin Nanovector recycling in podocytes mediated by FcRn; (**D**) interaction of secreted protein acidic and rich in cysteine (SPARC) favoring ANV accumulation in the extracellular space.

**Figure 4 biomolecules-09-00218-f004:**
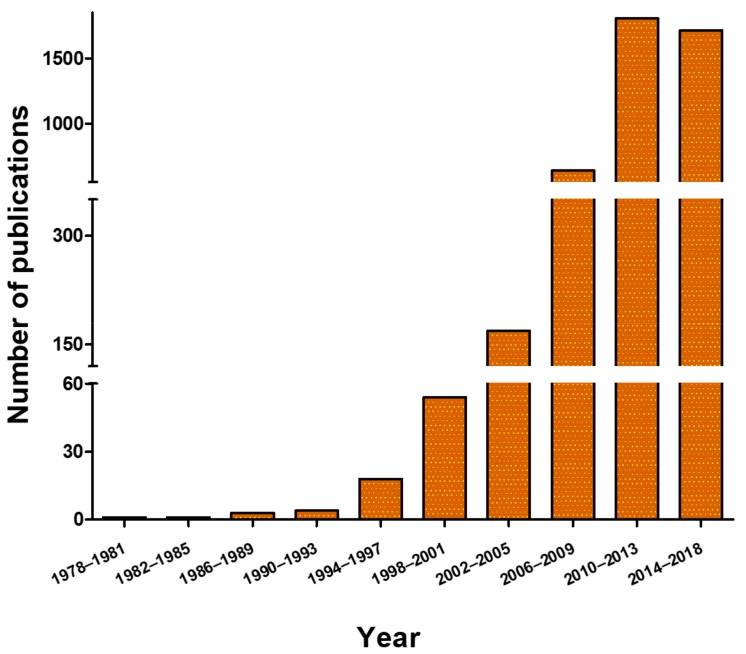
Number of scientific works based on ANVs published since 1978.

**Figure 5 biomolecules-09-00218-f005:**
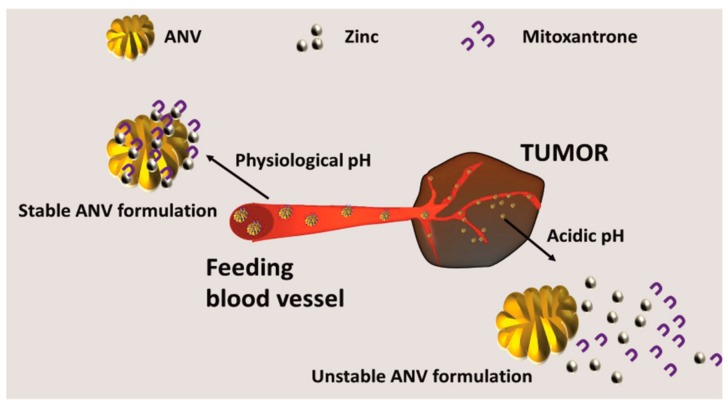
Generation of smart environmentally responsive ANVs, namely: pH-responsive ANVs were generated via the classical denaturation protocols. The system was doped with zinc to generate coordination bonds between the ANVs and the therapeutic mitoxantrone. This bond is stable at a physiologic pH, but at the acidic pH of the tumor microenvironment, it results in being unstable, allowing for the release of the therapeutic.

**Figure 6 biomolecules-09-00218-f006:**
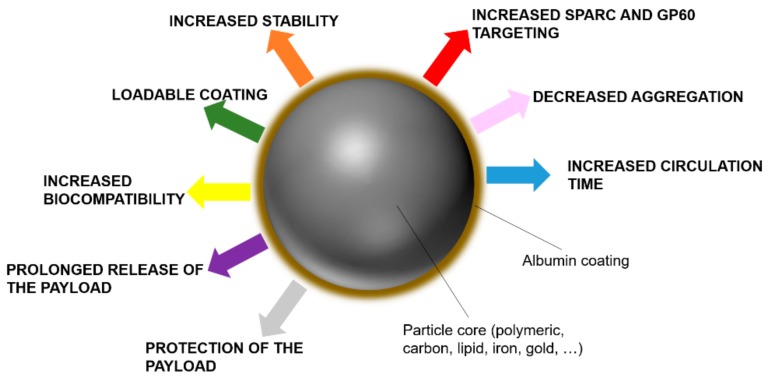
Benefits of albumin on the surface of nanocarriers.

**Table 1 biomolecules-09-00218-t001:** Active and completed clinical trials based on albumin nanovector (ANV) treatment for cancer.

Payload	Other Drugs	Disease	Phase	Status	ClinicalTrials.gov Identifier
PXT	None	Metastatic Breast Cancer	2	Completed	NCT00251472
PXT	None	Metastatic Breast Cancer	2	Active	NCT01463072
Rapamycin	None	Different Cancer with mTOR Mutations	1	Active	NCT02646319
PXT	Cetuximab Radiation Therapy	Head and Neck squamous cell carcinoma	1	Completed	NCT00736619
PXT	Carboplatin	Luminal B/HER-2 Negative Breast Cancer	4	Recruiting	NCT03799692
Rapamycin	Pazopanib	Sarcomas	1/2	Recruiting	NCT03660930
PXT	Gemcitabine and Bevacizumab	Metastatic Breast Cancer	2	Completed	NCT00662129
Rapamycin	Temozolomide and Irinotecan	Recurrent or Refractory Solid Pediatric Tumors	1	Recruiting	NCT02975882
PXT	Gemcitabine	Unresectable Pancreatic Cancer	1	Recruiting	NCT02336087
PXT	Sargramostim	Chemoresistant Tumors	2	Completed	NCT00466960
PXT	5-Fluorouracil, Epirubicin, and Cyclophosphamide	Breast Cancer	2	Completed	NCT00110695
PXT	Bevacizumab and Temozolomide	Unresectable Malignant Melanoma	2	Completed	NCT00626405
PXT	None	Peritoneal Neoplasms	1	Completed	NCT00666991

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
