# Peer review of "Albumin Nanovectors in Cancer Therapy and Imaging"

_biomolecules, 2019, doi:10.3390/biom9060218_

Reviewer 1 Report

This is a scholarly work and the authors have done a good job of reviewing this emerging field of albumin nanovectors and their application in cancer therapy. 

My main criticism is although the title saying imaging, the authors do not discuss the imaging as applied to cancer systems in details in the manuscript. I will love to see some specific examples of cancer imaging employing albumin nanovectors from the current literature.

Overall, I think this should be published with some revision.

The text need some English editing. There are some typos and sentence construction issues.

Figure 4 needs some formatting. There are some space issues in the text, particularly prominent where the figure was placed

Author Response

This is a scholarly work and the authors have done a good job of reviewing this emerging field of albumin nanovectors and their application in cancer therapy.

My main criticism is although the title saying imaging, the authors do not discuss the imaging as applied to cancer systems in details in the manuscript. I will love to see some specific examples of cancer imaging employing albumin nanovectors from the current literature.

We thank the reviewer for her/his efforts in reviewing our work. We implemented the manuscript with more examples dedicated to cancer imaging based on ANV, including the medical approved system nanocoll.

Overall, I think this should be published with some revision. The text need some English editing. There are some typos and sentence construction issues.

We did our best in improving the quality of the document and enhancing its readability.

Figure 4 needs some formatting. There are some space issues in the text, particularly prominent where the figure was placed

Figure 4 was reformatted.

Reviewer 2 Report

The review manuscript is about the preparation of albumin-based nanovectors as drug delivery systems and their use in cancer therapy and diagnosis. It explaines the synthesis pathways clearly and gives a lot of literature for their applicaton areas. The subtitles are in a reasonable flow and the pictures in between support the text. Only one comment: It might be much more clear in the introduction part if authors explain in couple of sentences what nanovector is or why albumin protein is used for generating nanovectors, or why albumin proteins are preferred in nanovector shape tob e used as nanocarriers for drug delivery applications.  

Author Response

We thank the reviewer for her/his nice words on our work. The word nanovector is used as a synonym of nanocarrier or nanoparticles. However, considering that ANV are not only used for drug delivery but also for diagnostic and/or theranostic purposes, we believe the word vector is more appropriate. A brief definition of nanovector was included and in the very first part of the paper, where we described why biological molecules, and albumin in particular, can be sometimes preferred to develop effective nanomedicine. 

Reviewer 3 Report

Attached document

Author Response

The review by Parodi et al., describes albumin nano-vectors as a carrier for cancer therapeutic agents. The authors begin with the structure of albumin followed by methods of synthesis of albumin-based nanoparticles and mechanism of cellular uptake of albumin nanoparticles. The authors further describe albumin-based nanocarriers as delivery agents for cancer therapeutic and imaging agents.

We thank the reviewer for her/his work, and we are confident that the provided comments will improve the quality of the paper.

Comments:

1)    The field of albumin-based nanocarriers has been reviewed extensively in literature. Thus, it is essential to discuss newer strategies in detail.  There is recent focus on devising strategies to synthesize nano-carrier system to harness the endogenous albumin following administration. Authors need to discuss these strategies.

Ex: DOI:10.1016/j.biomaterials.2018.06.002, DOI:10.1016/j.jconrel.2017.05.004, DOI:10.1021/jacs.8b12167.

A new paragraph named “Harnessing endogenous Albumin and hybrid systems” was included and all of these works were cited in addition to others in which the albumin coating is applied on nanomaterial in the synthesis phase. A new Figure for this chapter was added as well.

2)    How does exogenous albumin coating of nanocarriers effect the physiochemical characteristics of nanocarriers such as size, zeta-potential etc? Authors need to discuss these effects.

The effects of the albumin coating were discussed in the aforementioned section.

3)    How does route of delivery effect the efficacy of albumin-based nanocarriers? Additional section needs to be included, discussing studies which explore different route of delivery.

Many works are describing different administration route other than intravenous injection. However, only the inhalation was associated with cancer disease. For this topic, we included an additional brief section called inhalable ANV.

4)    Authors discuss multiple studies wherein albumin-based nanocarriers are exploited for delivery of small molecules as cancer therapy. However, albumin-based nanocarriers are also explored for delivery of oligonucleotides and other nucleic acids which needs to be reviewed.

Different papers describing this kind of biological pharmaceutics were included in the proper sections.

5)    There multiple clinical trials which use albumin-based nanocarriers for cancer therapy. A table describing these clinical trials needs to be included.

A table reporting terminated or ongoing trials based on ANV was included.

Round  2

Reviewer 3 Report

Authors have addressed all the queries.